# The Identification of Viral Pathogens in a *Physostegia virginiana* Plant Using High-Throughput RNA Sequencing

**DOI:** 10.3390/v15091972

**Published:** 2023-09-21

**Authors:** Jinxi Dong, Yuanling Chen, Yi Xie, Mengji Cao, Shuai Fu, Jianxiang Wu

**Affiliations:** 1Hainan Institute, Zhejiang University, Sanya 572025, China; 22016215@zju.edu.cn (J.D.); 22116216@zju.edu.cn (Y.C.); 2State Key Laboratory of Rice Biology, Key Laboratory of Biology of Crop Pathogens and Insects of Zhejiang Province, Institute of Biotechnology, Zhejiang University, Hangzhou 310058, China; 11616061@zju.edu.cn; 3National Citrus Engineering and Technology Research Center, Citrus Research Institute, Southwest University, Chongqing 400712, China; 4Research Center for Biological Computation, Zhejiang Lab, Hangzhou 311100, China

**Keywords:** *Physostegia virginiana*, RNA-seq, *Physostegia virginiana* crinkle-associated virus 1, *Physostegia virginiana* caulimovirus 1, *Physostegia virginiana* caulimovirus 2, *Physostegia virginiana* fijivirus

## Abstract

*Physostegia virginiana* is an important ornamental and cut-flower plant in China. Its commonly used method of clonal propagation leads to virus accumulation in this plant. However, which viruses can infect the *Physostegia virginiana* plant remains to be illuminated. In this work, five viral pathogens in a *Physostegia virginiana* plant with virus-like symptoms of yellow, shriveled, and curled leaves were identified using RNA-seq, bioinformatics, and molecular biological techniques. These techniques allowed us to identify five viruses comprising one known alfalfa mosaic virus (AMV) and four novel viruses. The novel viruses include a virus belonging to the genus *Fabavirus*, temporarily named *Physostegia virginiana* crinkle-associated virus 1 (PVCaV1); two viruses belonging to the genus *Caulimovirus*, temporarily named *Physostegia virginiana* caulimovirus 1 and 2 (PVCV1 and PVCV2); and a virus belonging to the genus *Fijivirus*, temporarily named *Physostegia virginiana* fijivirus (PVFV). The genome sequences of PVCaV1, PVCV1, and PVCV2, and the partial genome sequence of PVFV were identified. Genome organizations and genetic evolutionary relationships of all four novel viruses were analyzed. PVCaV1 has a relatively close evolutionary relationship with five analyzed fabiviruses. PVCV1 and PVCV2 have separately a closest evolutionary relationship with lamium leaf distortion-associated virus (LLDAV) and figwort mosaic virus (FMV), and PVFV has a close evolutionary relationship with the five analyzed fijiviruses. Additionally, PVCaV1 can infect *Nicotiana benthamiana* plants via friction inoculation. The findings enrich our understanding of *Physostegia virginiana* viruses and contribute to the prevention and control of *Physostegia virginiana* viral diseases.

## 1. Introduction

*Physostegia virginiana*, also known as false dragon head flower, belongs to the *Physostegia* genus of the *Lamiaceae* family and is a perennial herbaceous plant. *Physostegia virginiana* is an important ornamental plant in many public gardens and a cut-flower plant in China because of its beautiful and colorful flower; it was also one of the eight flowers presented to winners at the 2008 Beijing Olympic Games [1]. The clonal propagation method commonly used by *Physostegia virginiana* leads to the accumulation of a variety of viruses in this plant, which seriously affects the quality and ornamental effect of flowers. However, so far, alfalfa mosaic virus (AMV) is the only reported virus infecting *Physostegia virginiana* [2]; thus, what other viruses can infect the *Physostegia virginiana* plant remains to be known.

High-throughput sequencing technology, RNA sequencing (RNA-seq), can simultaneously sequence millions or even tens of millions of DNAs and has been widely used to identify novel viruses and enriches the virus resource in recent years [3]. Compared with the first-generation sequencing technology and conventional virus detection techniques such as serological enzyme-linked immunosorbent assay (ELISA) [4] and molecular polymerase chain reaction (PCR)/reverse transcription PCR (RT-PCR), it has obvious advantages [5,6]. For example, its throughput is very high, only a small number of samples are required, the genomic characteristic of the virus is not necessary to be known in advance, and the detection accuracy is up to a single base [7]. Using the RNA-seq technique, numerous new viruses were identified [8,9,10,11].

Fabaviruses belong to the genus *Fabavirus*, *Comovirinae* subfamily, *Secoviridae* family, and *Picornavirales* order, with a number of hosts, including dicotyledonous and some monocotyledonous plants [12]. Their virions are spherical, approximately 30 nm in diameter, and spread by aphids in a nonpersistent manner and through mechanical friction. Fabaviruses have a bipartite single-stranded positive-sense RNA genome consisting of RNA1 (6.0~6.3 kb) and RNA2 (3.6~4.5 kb) with a genome-linked protein (VPg) covalently attached at the 5′ end and a poly (A) tail at the 3′ end [13]. Each genomic RNA encodes a single polyprotein precursor, which is processed into functional proteins through proteolytic cleavage. The polyprotein encoded by RNA1 is processed into protease cofactors (Pro-Co), putative helicase (Hel), viral genome-linked protein (VPg), protease (Pro), and RNA-dependent RNA polymerase (RdRp) [14,15,16,17,18,19]. The polyprotein encoded by RNA2 is processed into viral movement protein (MP), large coat protein (LCP), and small coat protein (SCP) [20,21].

The genomes of caulimoviruses in the genus *Caulimovirus*, family *Caulimoviridae* of the order *Ortervirales*, are a 7.8~8.2 kb noncovalent closed circular double-stranded DNA (dsDNA), and its negative-sense chain has a discontinuous point, while the positive-sense chain has two or three discontinuous points (http://ictv.global/report/caulimoviridae, accessed on 10 July 2023). Caulimoviruses usually have 6~7 open reading frame (ORFs) encoding movement protein (MP), aphid transmission factor (ATF), DNA-binding protein (DNAb), coat protein (CP), polymerase polyprotein (PP), and inclusion body protein (IB) [22]. Previous studies have shown that some viral sequences of caulimoviruses can be integrated into the genome of host plants [23].

The genomes of fijiviruses are composed of 10 linear dsRNA segments 23~29 kb in size and characterized by low GC content (mostly 34–36%) [24]. Virions of fijiviruses are double-layered, spherical particles with 65–70 nm in diameter. Reported fijiviruses can be transmitted by planthoppers in a persistent and propagative manner. Each genomic segment contains conserved 5′ and 3′ terminal nucleotide sequences. Adjacent to the conserved terminal sequences, each genome segment possesses inverted repeats of several base pairs, similar to that of phytoreovirus and oryzavirus, although the sequences involved are different (https://ictv.global/report/chapter/spinareoviridae/spinareoviridae/fijivirus, accessed on 4 June 2023).

In 2021, during the surveys of viral diseases in the China National Botanical Garden in Beijing City, China, we found a *Physostegia virginiana* plant with virus-like symptoms of yellow, shriveled, and curled leaves, and collected it. Then, we used RNA-seq to uncover viral pathogens in this plant. Combined with RT-PCR, PCR, nucleic acid sequencing, and BLAST search, we identified a coinfection of alfalfa mosaic virus (AMV) and four novel viruses in this *Physostegia virginiana* plant. The four new viruses have been temporarily named *Physostegia virginiana* crinkle-associated virus 1 (PVCaV1), *Physostegia virginiana* caulimovirus 1 and 2 (PVCV1 and PVCV2), and *Physostegia virginiana* fijivirus (PVFV), respectively. Genome organizations and genetic evolutionary relationships of PVCaV1, PVCV1, PVCV2, and PVFV were analyzed.

## 2. Materials and Methods

### 2.1. Plant Samples and Plant Growth

In 2021, a *Physostegia virginiana* plant showing obvious yellow, shriveled, and curled leave symptoms was collected from the Beijing Botanical Garden in Beijing City, China, and stored in a laboratory refrigerator at −80 °C for further experiments. *N. benthamiana* plants were grown inside a growth chamber maintained at 25 °C (day)/18 °C (night), 60% relative humidity, and a 16 h (light)/8 h (dark) photoperiod.

### 2.2. RNA Sequencing (RNA-Seq) and De Novo Assembly

RNA-seq and de novo assembly were performed as previously described [11]. In brief, we extracted total RNA from the leaves of the collected *Physostegia virginiana* plant using an EASYspin Plus Complex Plant RNA Kit (Aidlab Biotech, Beijing, China) for subsequent RNA-seq. Ribosomal RNAs were removed using a TruSeq RNA Sample Prep Kit (Illumina, San Diego, CA, USA). An RNA library was constructed using the TruSeq RNA Sample Prep Kit (Illumina, San Diego, CA, USA) according to the manufacturer’s protocol, followed by sequencing on the Illumina HiSeq X-ten platform (Biomarker Technologies, Beijing, China). Low-quality reads were filtered, and adapters of the paired-end raw reads were trimmed using the CLC Genomics Workbench 9.5 software (QIAGEN, Hilden, Germany). The clean reads were assembled de novo into contigs using the Trinity v2.3.2 program with default parameters (Broad Institute and the Hebrew University of Jerusalem, Cambridge, Massachusetts and Jerusalem, USA, and Israel), and K-mer was 25 bp [10].

### 2.3. PCR, RT-PCR, and Rapid Amplification of cDNA Ends (RACE)

To identify the presence of possible viruses and obtain the genomes of novel viruses, we extracted RNA and DNA from the leaves of the collected *Physostegia virginiana* plant utilizing an RNAiso Plus reagent (TaKaRa, Tokyo, Japan) and Fast Pure Plant DNA Isolation Mini Kit (Vazyme, Nanjing, China). Using the reverse transcriptase M-MLV (TaKaRa, Kusatsu, Japan), total RNA was reverse-transcribed into cDNAs, followed by Super-Fidelity PCR using Phanta DNA polymerase (Vazyme, Nanjing, China) and specific detection primer pairs (Appendix A). The DNA/RNA sequences of viral contigs obtained via RNA-seq and assembly were PCR/RT-PCR-amplified using Super-Fidelity PCR with specific primer pairs (Appendix A). To amplify the 5′ and 3′ end sequences of the novel viruses, 5′/3′ RACE was performed using the HiScript-TS 5′/3′ RACE Kit (Vazyme, Nanjing, China) according to the manufacturer’s instructions. For DNA viruses with circular structures, we utilized Super-Fidelity PCR with specific primer pairs (Appendix A) to amplify the remaining unknown sequences outside the contigs. The PCR products were gel-purified and Sanger-sequenced, individually. The genomes of the four newly identified viruses were finally obtained from the diseased *Physostegia virginiana* plant through PCR, RT-PCR, and then DNA Sanger sequencing.

### 2.4. Analyses of Viral Genome and Proteins

ORFs in the viral genomic sequences were predicted using the ORFfinder service (http://www.ncbi.nlm.nih.gov/orffinder, accessed on 3 June 2022). The sixteen viral genome segments obtained in this study were used to perform BLASTx and BLASTp searches in the NCBI databases. Then, the conserved domains of the viral proteins were identified by using the conserved domain search service (CD-Search) in NCBI (https://www.ncbi.nlm.nih.gov/Structure/cdd/wrpsb.cgi, accessed on 5 June 2023) [25].

### 2.5. Phylogenetic Analysis

To illustrate the evolutionary relationships of these four new viruses with other viruses, phylogenetic trees based on the genomic sequences, CP, RdRp, and some functional proteins aa sequences were constructed. The phylogenetic analyses were performed using the neighbor-joining method in the MEGA X software (https://www.megasoftware.net/, accessed on 11 June 2022)with 1000 bootstrap replicates. The related viruses were downloaded from the GenBank database, and their detailed information is presented in Appendix A.

### 2.6. Mechanical Inoculation

Leaves of the collected *Physostegia virginiana* plant were ground to slurry in 0.1 M phosphate buffer (pH 7.2) for mechanical inoculation. After sprinkling a small amount of emery powder on the leaves of *N. benthamiana*, we dipped an index finger in the ground leaf sap and gently rubbed along the direction of the vein [10]. Finally, regular observation and photography were the methods used for recording the symptoms of the inoculated *N. benthamiana* plants.

## 3. Results

### 3.1. Identification of Five Viruses in Physostegia virginiana through RNA-Seq

A *Physostegia virginiana* plant showing yellow, shriveled, and curled leave symptoms (Figure 1A left) was collected for subsequent RNA-seq. The resulting 29,101,066 clean reads were assembled into contigs de novo. The assembled contigs longer than 1000 nucleotides (nt) were further subjected to BLASTn and BLASTx searches to identify potential viruses in the GenBank database. Then, 16 contig sequences with 1650, 1976, 2015, 2039, 2590, 2936, 2946, 3341, 3430, 3457, 3631, 3738, 4356, 5801, 7765, and 7887 nt (Appendix A) showed high homology with viral genome and were used for further analysis. Among these virus-like contig sequences, three contig sequences had high homology with the genomic RNA1-3 of AMV, two contig sequences had homology with genomic RNA 1 and RNA 2 sequences of viruses from the genus *Fabavirus*, two contig sequences had homology with the genome sequences of caulimoviruses, and nine contig sequences had homology with the genomic RNAs of fijiviruses (Appendix A). To confirm this RNA-seq result, we performed PCR and RT-PCR assays. The DNA or RNA sequences of specific fragments of the six contigs could be amplified in this plant (Figure 1B). PCR and Sanger sequencing of the PCR products further confirmed that this plant contained these 16 contig sequences separately from AMV and four novel viruses that were temporarily named *Physostegia virginiana* crinkle-associated virus 1 (PVCaV1), *Physostegia virginiana* caulimovirus 1 and 2 (PVCV1 and PVCV2), and *Physostegia virginiana* fijivirus (PVFV).

### 3.2. Determination of Genomic Sequences of the Four New Viruses

The designed primers, reverse transcriptase, and PCR high-fidelity polymerase were used to segmentally amplify the DNA/RNA sequences of the four new viral contig sequences. The PCR products connected to the cloning vector were sent to the company for Sanger sequencing, and all thirteen assembled contig sequences from the four novel viruses were verified. Then, 5′/3′ RACEs combined with Sanger sequencing were performed to obtain the 5′and 3′ terminal genomic sequences of PVCaV1. Genomic RNA1 and RNA2 of PVCaV1 contain, respectively, 5815 nt and 3297 nt, and have been deposited in the GenBank under the accession number OR208176 and OR208177. Full-length genomes of two novel circular DNA viruses, PVCV1 and PVCV2, were obtained using reverse PCR amplification and Sanger sequencing, and they have also been deposited in the GenBank under the accession number OR208178 and OR208179. Full-length genomes of PVCV1 and PVCV2 contain, respectively, 7765 nt and 7887 nt. The genomes of all known fijiviruses are composed of 10 dsRNA strands, but only contig-20685 from PVFV RNA1 (4424 nt, accession no. OR378266), contig-5043 from PVFV RNA2 (3742 nt, accession no. OR378267), contig-3081 from PVFV RNA3 (3470 nt, accession no. OR378271), contig-8459 from PVFV RNA4 (3674 nt, accession no. OR378268), contig-36481 from PVFV RNA5 (3012 nt, accession no. OR378269), contig-2375 from PVFV RNA7 (1992 nt, accession no. OR378272), contig-7856 from PVFV RNA8 (2948 nt, accession no. OR378273), contig-33929 from PVFV RNA9 (1691 nt, accession no. OR378270), and contig-25766 from PVFV RNA10 (2046 nt, accession no. OR378274) were detected in this study. The sequence information of PVFV RNA 6 is currently missing likely because of low virus abundance.

### 3.3. Genomic Structure and Phylogenetic Analysis of PVCaV1

PVCaV1 is a bicomponent virus containing RNA1 and RNA2, which has the typical genomic structure of fabaviruses (Figure 2A) [13]. PVCaV1 RNA1 is composed of 5815 nt with a 5′ untranslated region (5′ UTR) of 176 nt and a 3′ UTR of 74 nt, excluding the polyA tail (Figure 2A). This RNA1 contains a single ORF of 5565 nt that starts at nt position 177 and ends at nt position 5741, and encodes a 207.9 kDa polyprotein (named P1) that was speculated to be processed into Co-Pro (36.3 kDa), Hel (64 kDa), VPg (3 kDa), Pro (22.8 kDa), and Pol (77.2 kDa) based on similar cleavage sites with other fabaviruses (Figure 2A) [26,27,28,29,30,31]. This polyprotein shares 36.64~46.97% amino acid (aa) sequence identities with the eight known fabaviruses (Table 1). According to the results of CDD analysis, this polyprotein includes three conserved domains: a helicase domain (pfam00910, aa 495–601), a 3C-cysteine protease domain (pfam00548, aa 1067–1134), and an RdRp domain (pfam00680, aa 1329–1621) (Figure 2A). PVCaV1 RNA2 consists of 3297 nt with a 5′ UTR of 133 nt and a slightly shorter 3′ UTR (66 nt), and it contains a single ORF that begins at nt position 134 and ends at nt position 3231. This ORF encodes a polyprotein of 115.6 kDa (named P2) that was predicted to be cleaved into MP (47.5 kDa), LCP (44 kDa), and SCP (22 kDa) according to similar cleavage sites with other fabaviruses [32,33]. This polyprotein has a conserved protein domain, Como_LCP superfamily (pfam02247, aa 455–802), and shares 23.89~44.99% aa sequence identity with that of the eight known fabaviruses (Table 1). Based on the species discrimination criteria in the Fabavirus genus, the aa identities of CP and RdRp are, respectively, less than 75% and 80%, and the aa homology comparison results (Table 1) indicate that PVCaV1 is a new species of Fabavirus.

The P1 and P2 protein sequences of eight representative members of the genus Fabavirus and nine representative members of other genera in the family *Secoviridae* were downloaded from the GenBank database for phylogenetic tree construction (Appendix A). The phylogenetic trees showed that PVCaV1 and fabaviruses clustered in the same branch, which clarified its position in the phylogenetic tree, indicating that PVCaV1 belongs to fabaviruses (Figure 2B,C). Subsequently, the phylogenetic trees of PVCaV1 and the eight known viruses in the Fabavirus genus based on RdRp, LCP, and SCP also supported the above conclusion, and the evolutionary relationship between PVCaV1 and broad bean wilt virus 1 (BBWV1), broad bean wilt virus 2 (BBWV2), cucurbit mild mosaic virus (CuMMV), lamium mild mosaic virus (LMMV), and gentian mosaic virus (GeMV) was relatively close (Appendix A).

### 3.4. Genomic Structure and Phylogenetic Analyses of PVCV1 and PVCV2

Based on the results of RNA-seq, bioinformatic analysis, PCR, and Sanger sequencing, the genomes of PVCV1 and PVCV2 belonging to the *Caulimovirus* genus were detected. PVCV1 and PVCV2 have monopartite noncovalently closed circular dsDNA genomes with 7765 nt and 7887 nt. Each genome of PVCV1 and PVCV2 was predicted to contain six ORFs, namely MP, aphid transmission factor protein (ATF), DNA-binding protein (DNAb), CP, polymerase polyprotein (PP), and inclusion body protein (IB) (Figure 3A,B) [34]. According to the species classification criteria of the *Caulimovirus* genus, the aa identity of the reverse transcriptase is less than 80% [35], and based on the aa homology comparison results of PVCV1 and PVCV2 with 12 known caulimoviruses (Table 2), PVCV1 and PVCV2 are presumed to be two new species of the genus *Caulimovirus*.

The genome of PVCV1 began with the conserved tRNAMet sequence (5′-TGGTATCAGAGCC-3′), which served as a primer binding site and was complementary to the plant consensus sequence of the tRNAMet binding site. BLASTn search showed that the genome of PVCV1 had the highest homology of 48.55% with lamium leaf distortion associated virus (LLDAV, calimovirus) (Table 2). ORF1 of PVCV1 encodes the viral MP (nt 14-979). BLASTp alignment showed that the aa homology between PVCV1 MP and 12 representative caulimoviruses was 28.67~72.19%, and PVCV1 MP shared the highest homology (72.19%) with LLDAV (Table 2). A conserved domain MP superfamily (pfam01107, aa 54–232) was observed in PVCV1 MP (Figure 3B), including conserved motif A (GNLAYGKFMFT, aa 164–174) and motif B (GYALSNSHHSI, aa 218–228) (Figure 3B) [36]. In addition, the MP contained two conserved DXR motifs (aa 113–115 and 148–150) that may be functionally important [37]. ORF2 of PVCV1 (981–1508 nt) encodes a 19.8 kDa ATF protein, which contained the conserved domain Cauli_AT superfamily (pfam03233, aa 1–174) associated with aphid transmission (Figure 3B). The ATF shared the highest aa homology of 73.71% with LLDAV and aa homology of 23.19~60.13% with the 12 caulimoviruses (Table 2). It has been reported that the IXG motif (X refers to any amino acid) in caulimovirus ATF is a key site for the interaction between ATF and virus particles during aphid transmission [38], and the ICG motif was found in PVCV1 ATF (aa 53–55). ORF3 of PVCV1 (1505–1738 nt) encodes an 8.9 kDa DNAb, which is a DNA-binding protein containing the Cauli_DNA-bind superfamily conserved domain (pfam03310, aa 5–63) (Figure 3B). PVCV1 DNAb also had the highest aa homology with LLDAV (52.90%). DNAb is an important factor in aphid transmission, interacting with CP, MP, and ATF [39]. ORF4 of PVCV1 was found to encode a CP, which shared the highest aa identity (36.38%) with that of soybean putnam virus (SPuV) (Table 2) and contained the conserved zinc finger motif (CX2CX4HX4C, aa 414–430) (Figure 3B). ORF5 of PVCV1 (3291–5282 nt) was predicted to encode a putative PP with 77.1 kDa. CDD online analysis results indicate that PP has three conserved domains: peptidase_A3 superfamily (pfam02160, aa 13–215) containing the conserved peptidase motif LHCYVDTGASLC (aa 33–44); reverse transcriptase RT_LTR domain (pfam00078, aa 264–443), containing the conserved reverse transcriptase motif VYVDDILVYS (aa 391–400); and RNase H domain (RNase_HI_RT_Ty3, pfam17917, aa 538–659), containing the conserved motif FWGGVLKA (aa 547–554) of RNase H (Figure 3B) [40]. PVCV1 PP shared the highest aa identity (68.28%) with that of cauliflower mosaic virus (CaMV) (Table 2). ORF6 of PVCV1 (5458–7011 nt) was predicted to encode a putative inclusion body protein (IB) (517 aa, 59.0 kDa), which was a major component of viral inclusion bodies [41]. Viral inclusion bodies are the site of virus assembly, DNA synthesis, and accumulation. PVCV1 IB shared the highest aa identity (71.19%) with that of LLDAV (Table 2) and contained the conserved motif GLINTIYP (aa 316–323) and the conserved domain Cauli_VI (pfam01693, aa 169–209) (Figure 3B).

In addition, the genome of PVCV1 contains a large intergenic region of 767 bp between ORF6 and ORF1, and a small intergenic region of 175 bp between ORF6 and ORF5. An RNA polyadenylation signal (AATAA, nt 7262–7266) and a negative-sense strand primer-binding site sequence (5′-TGGTATCAGAGCC-3′, nt 1–13) are present in the large intergenic region, while a TATA box (5′-TATAAATA-3′, nt 5353–5360) is present in the small intergenic region, as published previously [42,43].

The genome of PVCV2 began with the conserved tRNAMet sequence 5′-TGGTATCAGAGC-3′, which was a primer binding site and complementary to the consensus sequence of plant tRNAMet binding sites. BLASTn search showed that the genome of PVCV2 shared the highest nt identity (44.48%) with that of figwort mosaic virus (FMV) (Table 3). ORF1 of PVCV2 (13–993 nt) was predicted to encode 37.0 kDa viral MP as PVCV1 ORF1 by the ORF Finder soft (https://www.ncbi.nlm.nih.gov/orffinder/, accessed on 3 August 2022). BLASTp alignment showed that PVCV2 MP shared the highest aa identity of 57.84% with that of FMV (Table 3). This MP contained MP superfamily conserved domain (pfam01107, aa 32–231), and two conserved motifs, motif A (GNLAYGKFIFT, aa 159–169) and motif B (GYALSNSHHSI, aa 213–223) (Figure 3C). ORF2 of PVCV2 (986–1507 nt) was predicted to encode a 19.9 kDa ATF, which shared the highest aa identity of 55.51% with FMV, and 19.23~43.89% aa identity with the 12 viruses from the *Caulimovirus* genus (Table 3). CDD analysis revealed that the Cauli_AT superfamily domain (pfam03233, aa 10–173), related to aphid virus transmission, was present in PVCV2 ATF (Figure 3C). Previous studies have revealed that the IXG motif in ATFs of caulimoviruses is critical for the interaction between ATF and virus particles during aphid virus transmission [44]. However, PVCV2 ATF does not contain an IXG motif. ORF3 of PVCV2 (1504–1854 nt) was predicted to encode a 12.8 kDa DNAb, which also contained the Cauli_DNA-bind superfamily domain (pfam03310, aa 2–111) (Figure 3C). PVCV2 DNAb also shared the highest aa identity of 54.16% with that of FMV (Table 3). ORF4 of PVCV2 (1852–3356 nt) was predicted to encode 58.1 kDa CP, which shared the highest aa identity of 46.10% with that of FMV (Table 3). Like PVCV1, PVCV2 CP also contained a conserved zinc finger motif (CX2CX4HX4C, aa 418-428) (Figure 3B,C). ORF5 of PVCV2 (3277–5316 nt) was predicted to encode a 79.1 kDa putative PP, which shared the highest aa identity of 68.15% with that of FMV (Table 3). Like PVCV1, PVCV2 PP also contained three conserved domains: peptidase_A3 superfamily (pfam02160, aa 55–262), including the conserved peptidase motif (IHCYVDTGASLC, aa 75–86); the RT_LTR domain of reverse transcriptase (pfam00078, aa 292–471), including the conserved reverse transcriptase motif (VYVDDIIVFS, aa 419–428); and the RNase H domain (RNase_HI_RT_Ty3, pfam17917, aa 566–679), including the conserved RNase H motif (DWGCVLKA, aa 575–582) (Figure 3C). ORF6 of PVCV2 (5348–7042 nt) was predicted to encode a 63.8 kDa putative IB, which also contained the conserved motif GLTRMIYP (aa 345–352) of the inclusion body protein (Figure 3C) and shared the highest aa identity of 40.79%with that of FMV (Table 3).

In addition, the genome of PVCV2 contained a large intergenic region of 857 bp between ORF6 and ORF1 and a small intergenic region of 31 bp between ORF6 and ORF5. An RNA polyadenylation signal (AATAA, nt 7262–7266) and a negative-sense strand primer-binding site sequence (5′-TGGTATCAGAGC-3′, nt 1–12) were present in the large intergenic region, as published previously [43].

Genome sequences of 11 known viruses in the genus *Caulimovirus*, as well as 14 representative members from the family *Caulimoviridae* in the GenBank database (Appendix A), were used for the phylogenetic tree construction of PVCV1 and PVCV2. Phylogenetic analysis was performed using MEGA version 10.0 and the neighbor-joining method. The phylogenetic tree based on the virus’s full-length genomic sequence indicated that PVCV1 and PVCV2 were clustered on a branch with caulimoviruses (Figure 4). PVCV1 had the closest genetic relationship with LLDAV, while PVCV2 had the closest genetic relationship with FMV. The phylogenetic tree based on PP, MP, CP, ATF, DNAb, and IB aa sequences of PVCV1, PVCV2, and 11 other viruses from the genus *Caulimovirus* also supports the above conclusion (Appendix A).

### 3.5. Genomic Structure and Phylogenetic Analyses of PVFV

ORF1 of PVFV RNA1 was predicted to encode a RdRp of 168.5 kDa (hereafter designated P1) that shared the highest aa sequence identity (45.90%) with the P1 of southern rice black-streaked dwarf virus (SRBSDV) (Table 4). According to the classification criterion of the genus *Fijivirus*, the aa identity of RdRp is less than 80%, and thus we assumed that PVFV is a new species of the genus *Fijivirus*. ORF2 of PVFV RNA2 was predicted to encode a putative protein of 130.9 kDa that shared the highest aa sequence identity (35.17%) with the P2 of maize rough dwarf virus (MRDV) (Table 4). PVFV RNA3 sequence currently obtained was incomplete, and partial ORF3 was found to share the highest aa sequence identity (32.56%) with the P3 of Fiji disease virus (FDV) (Table 4). A putative protein of 112.3 kDa was encoded by ORF4 of PVFV RNA 4 and shared the highest aa sequence identity (21.70%) with the P4 of rice black-streaked dwarf virus (RBSDV) (Table 4). ORF5 of PVFV RNA 5 was predicted to encode a putative protein of 101.1 kDa that shared the highest aa sequence identity (25.18%) with the P5 of MRDV (Table 4). The obtained PVFV RNA7 encoded two proteins, namely a complete protein P7-1 and a part of P7-2. P7-1 shared the highest aa sequence identity (29.26%) with that of Mal de Rio Cuarto virus (MRCV), while the obtained partial P7-2 shared the highest aa sequence identity (5.37%) with that of SRBSDV. The obtained ORF8 of PVFV RNA8 contained a conserved domain PTP_DSP_cys superfamily (pfam00782) that was incomplete and shared the highest aa sequence identity (20.93%) with the P8 of FDV (Table 4). PVFV RNA9 was predicted to encode two complete proteins P9-1 and P9-2 that shared the highest aa sequence identity (26.42% and 19.84%) with that of MRDV (Table 4). The obtained partial PVFV RNA 10 was found to encode an incomplete protein P10 that contained a conserved domain, namely Fiji_64_capsid superfamily (pfam05880), and shared the highest aa sequence identity (21.69%) with that of RBSDV (Table 4).

To illustrate the evolutionary relationship of PVFV with known viruses, a phylogenetic tree based on the RdRp aa sequences of PVFV, 24 representative viruses from different genera of the family *Spinareoviridae*, and tobacco mosaic virus (TMV) of the genus *Tobamovirus* from the NCBI database (Appendix A) was constructed. The resulting phylogenetic tree indicated that PVFV was clustered in a clade together with SRBSDV, MRCV, MRDV, FDV, and RBSDV from the genus *Fijivirus* (Figure 5).

### 3.6. Mechanical Inoculation of PVCaV1

The leaves from the collected *Physostegia virginiana* plant were homogenized and used to mechanically inoculate the leaves of *Nicotiana benthamiana* plants. The inoculation experiments consisted of three biological replicates with each performed on five *N. benthamiana* plants. However, there were no obvious virus-like symptoms on inoculated plants (Figure 6A). In order to determine the infection of the five viruses identified in this study, RT-PCR and PCR were performed. According to the results of RT-PCR and PCR, only PVCaV1 infection was detected in all inoculated plants (Figure 6B), and infection with the other four viruses was not detected, indicating that PVCaV1 can infect *N. benthamiana* plants through mechanical inoculation.

## 4. Discussion

In this work, we used RNA-seq, bioinformatics, and molecular biological techniques to investigate the viral pathogens of a *Physostegia virginiana* plant with symptoms of yellow, shriveled, and curled leaves and identified AMV and four novel viruses infecting this plant. The four novel viruses include a new virus belonging to the Fabivirus genus, provisionally designated *Physostegia virginiana* crinkle-associated virus 1 (PVCaV1); two new viruses belonging to the genus *Caulimovirus*, provisionally designated *Physostegia virginiana* caulimovirus 1 and 2 (PVCV1 and PVCV2); and a new virus belonging to the genus *Fijivirus*, provisionally designated *Physostegia virginiana* fijivirus (PVFV). The full-length genome sequences and genome organizations of PVCaV1, PVCV1, and PVCV2 and most genome sequences and genome organization of the nine RNA segments of PVFV were identified. This is the first time that fabavirus, caulimovirus, and fijivirus were identified in *Physostegia virginiana* plants. The results of the phylogenetic tree and viral protein aa identity analyses show that PVCaV1 is clustered in the branch of fabaviruses and has a close evolutionary relationship with BBWV1, BBWV2, CuMMV, LMMV, and GeMV (Figure 2 and Appendix A; Table 1). PVCV1 and PVCV2 are clustered in the branch of caulimoviruses, and PVCV1 has the closest evolutionary association with LLDAV, while PVCV2 has the closest evolutionary association with FMV (Figure 4; Table 2 and Table 3). PVFV is clustered in the branch of fijiviruses and has a close evolutionary relationship with SRBSDV, MRCV, MRDV, FDV, and RBSDV (Figure 5; Table 4). The findings presented in this study enrich our understanding of *Physostegia virginiana* viruses and contribute to effective strategies for the prevention and control of the viral diseases of *Physostegia virginiana*.

Among these five identified viruses, only PVCaV1 can infect *N. benthamiana* plants through friction inoculation, but it cannot cause obvious virus-like symptoms in the infected *N. benthamiana* plants (Figure 6A). Therefore, which virus or viruses cause yellow, shriveled, and curled leaf symptoms in the *Physostegia virginiana* plant needs to be further studied.

The propagation methods of *Physostegia virginiana* plants include sowing, ramet, and cottage, whereas the ramet and cottage are the main propagation methods of *Physostegia virginiana* plants because of their low seed setting rate and fewer seeds. The frequently used clonal propagation method may lead to the accumulation of a variety of viruses in *Physostegia virginiana* plants. Thus, we believe that the phenomenon of multiple virus coinfection in *Physostegia virginiana* plants is closely related to its horticultural propagation mode. In the future, it is necessary to strengthen the virus quarantine, keep virus-free plants for reproduction, and control transmission vectors such as aphids and leafhoppers.

Whether PVCaV1, PVCV1, PVCV2, and PVFV are transmitted by insect vectors in the field needs to be further verified. However, according to the published reports about viruses in the same genus [45,46,47], we speculate that PVCV1, PVCV2, and PVCaV1 may be transmitted by aphids, and PVFV may be transmitted by planthoppers. It has been reported that some viruses from the *Caulimovirus* genus can integrate their partial genome sequence into the genomes of their hosts [48]. Whether PVCV1 and PVCV2 can integrate their genomes into the host *Physostegia virginiana* plants remains to be fully explored using Southern blot hybridization and other molecular experiments. Furthermore, the full-length genome including RNA 6 and the genomic structure of PVFV need to be further determined, and the possible synergistic or antagonistic effects between these viruses need to be further elucidated. In addition, a virus field survey needs to be performed to clarify the prevalence of the five identified viruses in *Physostegia virginiana* plants in China and their host ranges, and the potential threats of these four new viruses to crops need to be assessed.

## Figures and Tables

**Figure 1 viruses-15-01972-f001:**
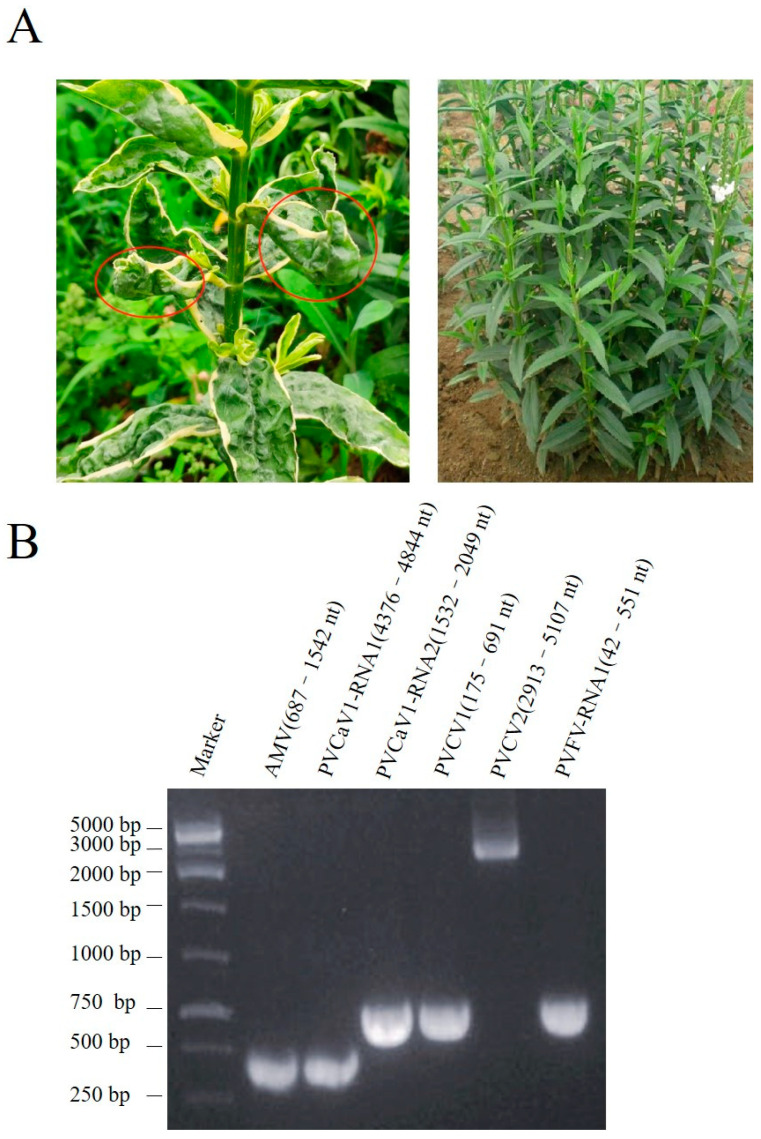
Symptoms of the collected *Physostegia virginiana* plant (**A**) and PCR/RT-PCR detection results of five viral pathogens in the collected *Physostegia virginiana* plant (**B**): (**A**) the left panel shows a *Physostegia virginiana* plant with yellow, shriveled, and curled leaves (**left**) and the right panel shows a healthy *Physostegia virginiana* plant (**right**); (**B**) PCR/RT-PCR detection results of five viral pathogens in the collected *Physostegia virginiana* plant using the specific primer pairs designed according to six assembled long contig sequences.

**Figure 2 viruses-15-01972-f002:**
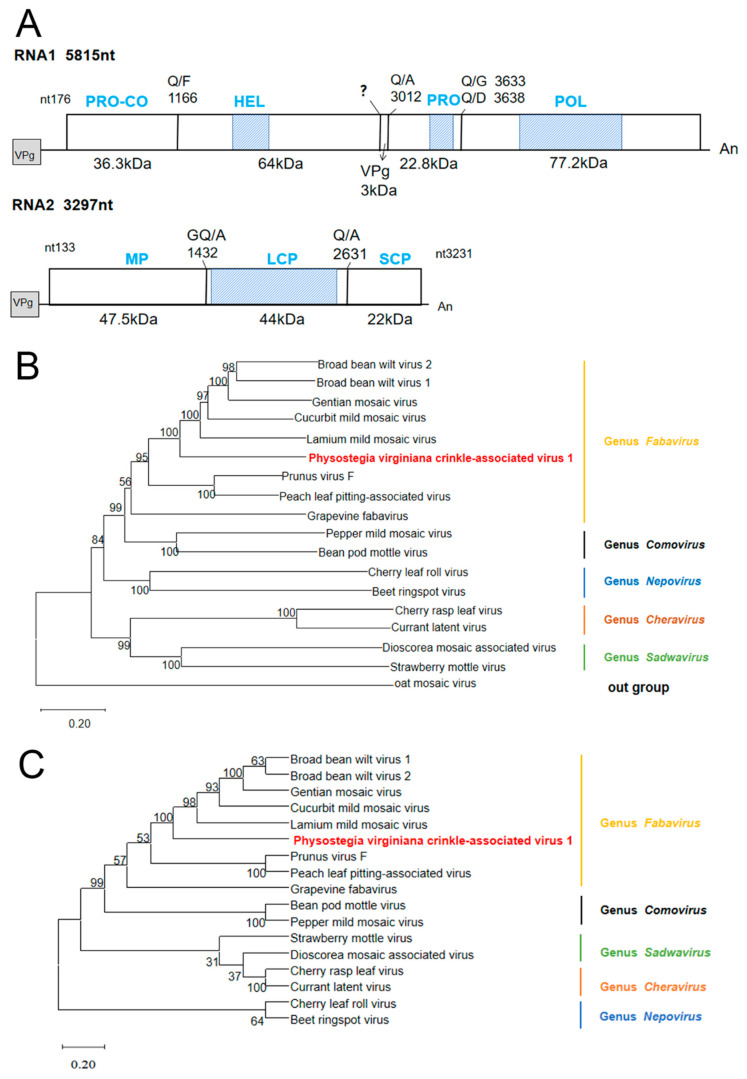
Genome organization and phylogenetic tree analysis of PVCaV1: (**A**) genome organization of PVCaV1. Horizontal lines indicate untranslated regions. Bars represent the polyprotein encoded by long open reading frames (ORFs). The predicted protease cleavage sites are indicated by vertical lines with the corresponding amino acid sites. “An” represents the polyadenylated tail; (**B**) the phylogenetic tree of PVCaV1 and 17 other viruses in the family *Secoviridae* based on the P1 protein; (**C**) the phylogenetic tree of PVCaV1 and 16 viruses in the family *Secoviridae* based on the P2 protein. The phylogenetic tree is constructed using the neighbor-joining method with 1000 bootstraps. Viruses from the same genus are shown in the same color. PVCaV is shown in red. The bootstrap values are indicated adjacent to the nodes. The accession numbers of these sequences are listed in Appendix A.

**Figure 3 viruses-15-01972-f003:**
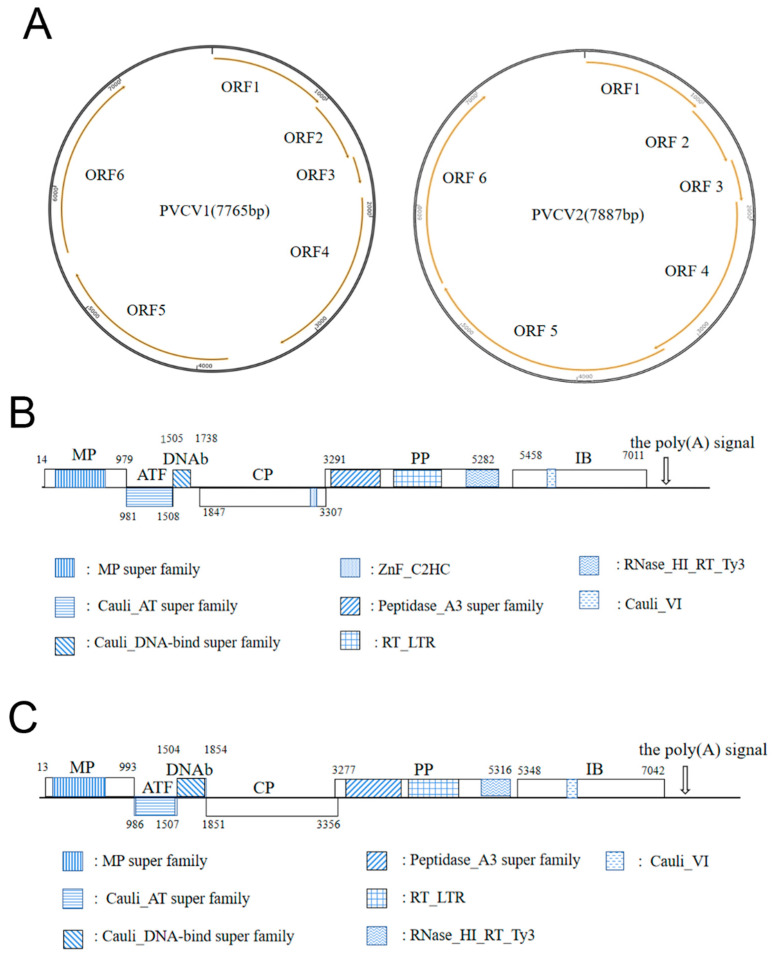
Genome diagrams of PVCV1 and PVCV2: (**A**) the left and right image separately shows the ring diagrams of PVCV1 and PVCV2 genomes. The orange lines with arrows represent each ORF; (**B**) the linear diagram of the PVCV1 genome. Each shaded part is a conserved domain; (**C**) the linear diagram of the PVCV2 genome. Each shaded part is a conserved domain.

**Figure 4 viruses-15-01972-f004:**
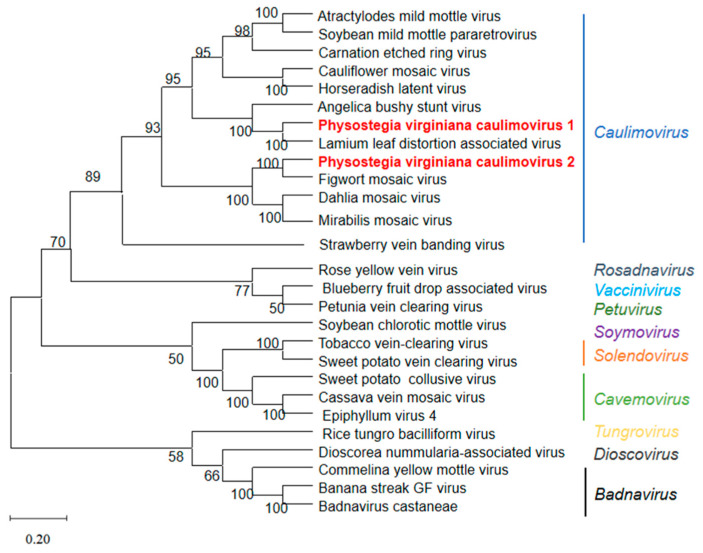
The phylogenetic tree of PVCV1, PVCV2, and other 25 known viruses from the family *Caulimoviridae* based on viral genome sequences. The phylogenetic tree was constructed using the neighbor-joining method with 1000 bootstraps. The bootstrap values are indicated adjacent to the nodes. Accession numbers of these sequences are listed in Appendix A.

**Figure 5 viruses-15-01972-f005:**
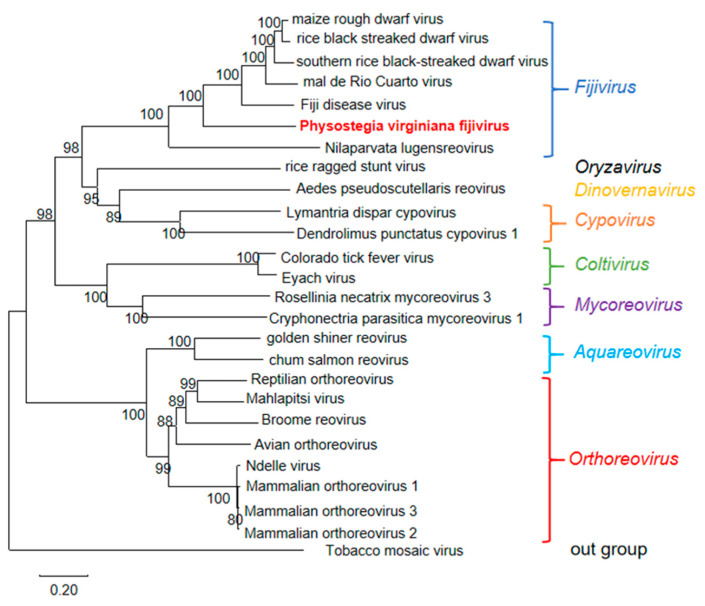
The phylogenetic tree based on RdRp amino acid sequences of PVFV, 24 representative viruses from different genera of the family *Spinareoviridae*, and tobacco mosaic virus (TMV) of the genus *Tobamovirus*. The phylogenetic tree was constructed using the neighbor-joining method with 1000 bootstraps. PVFV is shown in red. The bootstrap values are indicated adjacent to the nodes. Accession numbers of these sequences are listed in Appendix A.

**Figure 6 viruses-15-01972-f006:**
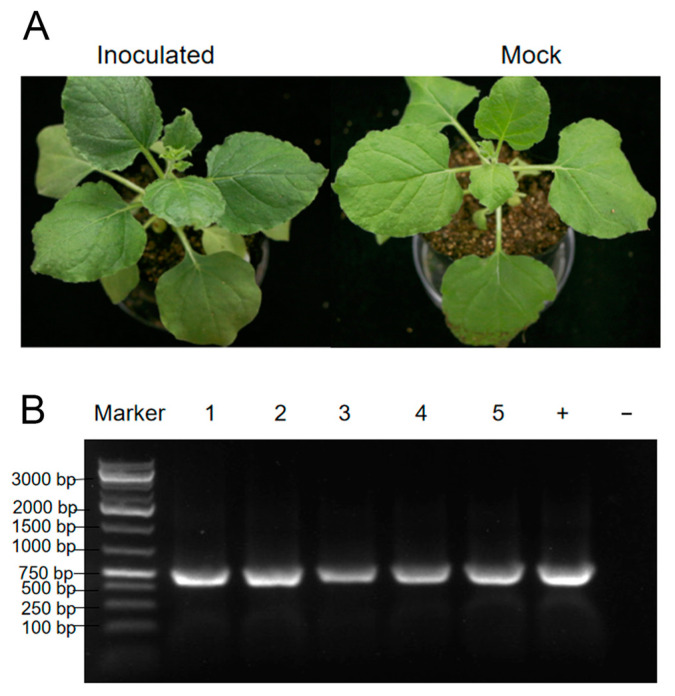
Symptoms of an *N. benthamiana* plant friction-inoculated with PVCaV1 (**A**) and the detection result of PVCaV1 infection in five inoculated plants using RT-PCR (**B**). Lanes 1–5 indicate the five inoculated *N. benthamiana* plants; lane ‘+’ is the collected *Physostegia virginiana* plant studied in this study, used as a positive control, and lane ‘−’ represents uninoculated *N. benthamiana* plant, used as a negative control.

**Table 1 viruses-15-01972-t001:** Viral protein aa sequence similarities between PVCaV1 and the other 8 members from the genus *Fabavirus*.

Virus Name	Amino Acid Identity (%)				
	P1	P2	RdRp	LCP	SCP
Broad bean wilt virus 1	44.83	44.99 ^1^	51.15	46.51 ^1^	41.06
Broad bean wilt virus 2	43.77	43.01	49.22	45.90	40.00
Cucurbit mild mosaic virus	46.97 ^1^	43.23	52.57 ^1^	46.39	43.54 ^1^
Peach leaf pitting-associated virus	43.87	25.33	48.79	26.96	19.77
Gentian mosaic virus	44.49	40.21	50.50	45.41	40.29
Grapevine fabavirus	36.64	24.36	45.43	20.52	17.49
Lamium mild mosaic virus	45.76	40.02	50.56	45.30	40.28
Prunus virus F	42.33	23.89	50.72	23.73	18.86
Broad bean wilt virus 1	44.83	44.99 ^1^	51.15	46.51 ^1^	41.06
Broad bean wilt virus 2	43.77	43.01	49.22	45.90	40.00

^1^ Red number indicates the highest similarity.

**Table 2 viruses-15-01972-t002:** Nucleotide and amino acid identities between PVCV1 and 12 viruses from the genus *Caulimovirus*.

Virus Name	Nucleotide Identities (%)	Amino Acid Identity (%)					
		MP (P1)	ATF (P2)	DNAb (P3)	CP (P4)	PP (P5)	IB (P6)
LLDAV	48.55 ^1^	72.19 ^1^	73.71 ^1^	52.90 ^1^	32.43	66.92	71.19 ^1^
CaMV	30.44	56.44	58.03	30.84	34.58	68.28 ^1^	32.72
HLV	24.85	55.04	60.13	37.41	36.13	65.05	32.58
FMV	23.32	44.30	34.61	30.43	35.07	65.43	18.16
PVCV2	18.10	43.89	32.57	27.05	30.82	60.47	21.18
AMMV	24.67	46.70	38.16	38.79	30.17	63.29	28.23
DaMV	18.90	45.65	32.09	18.00	31.82	50.21	19.47
MMV	24.05	45.12	23.19	23.88	25.68	61.44	21.59
CERV	26.68	44.82	46.12	34.00	36.00	64.81	30.64
SPuV	24.54	45.38	38.16	33.65	36.38 ^1^	64.09	25.94
AnBSV	25.19	42.23	41.54	30.43	31.27	66.30	27.91
SVBV	11.01	28.67	23.19	21.11	22.67	52.80	16.36

^1^ Red number indicates the highest identity.

**Table 3 viruses-15-01972-t003:** Nucleotide and amino acid identities between PVCV2 and 12 representative viruses from the genus *Caulimovirus*.

Virus Name	Nucleotide Identities (%)	Amino Acid Identity (%)					
		MP (P1)	ATF (P2)	DNAb (P3)	CP (P4)	PP (P5)	IB (P6)
FMV	44.48 ^1^	57.84 ^1^	55.51 ^1^	54.16 ^1^	46.10 ^1^	68.15 ^1^	40.79 ^1^
CaMV	13.00	51.50	31.41	30.85	29.80	56.03	19.71
MMV	16.27	48.83	41.87	38.02	36.90	59.86	27.19
HLV	14.52	45.71	29.27	33.21	31.08	55.26	20.49
AMMV	9.59	49.00	38.74	37.93	27.10	56.65	19.34
DaMV	18.20	45.03	38.15	44.17	35.97	53.53	27.65
SPuV	15.96	47.38	40.51	35.70	32.76	58.34	20.79
LLDAV	15.45	42.52	28.80	37.13	27.48	56.21	20.28
PVCV1	18.10	43.89	32.57	27.05	30.82	60.47	21.18
CERV	5.74	45.47	29.77	32.09	28.84	55.92	21.03
AnBSV	15.55	37.80	33.92	29.23	24.83	55.51	16.96
SVBV	6.54	33.18	19.23	None ^2^	24.38	46.51	14.50

^1^ Red number indicates the highest identity. ^2^ None, no submitted sequence.

**Table 4 viruses-15-01972-t004:** Viral protein amino acid sequence similarities between PVFV and the five known members in the genus *Fijivirus*.

Virus Name	Amino Acid Identity (%)										
	P1	P2	P3	P4	P5	P7-1	P7-2	P8	P9-1	P9-2	P10
MRDV	45.06	35.17 ^1^	32.27	21.63	25.18 ^1^	25.04	4.74	18.92	26.42 ^1^	19.84 ^1^	20.14
RBSDV	43.97	35.00	32.24	21.70	13.68	25.39	None ^2^	19.34	25.25	19.50	21.69 ^1^
SRBSDV	45.90 ^1^	34.97	32.53	19.38	21.99	25.23	5.37 ^1^	18.08	24.75	18.20	20.72
FDV	44.74	34.96	32.56 ^1^	19.71	19.87	24.17	None ^2^	20.93 ^1^	21.71	19.50	18.06
MRCV	45.41	35.08	None ^2^	19.51	16.42	29.26 ^1^	None ^2^	18.18	None ^2^	17.81	20.86

^1^ Red number indicates the highest similarity. ^2^ None, no submitted sequence.

## Data Availability

Not applicable.

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
