# Peer review of "The Identification of Viral Pathogens in a Physostegia virginiana Plant Using High-Throughput RNA Sequencing"

_viruses, 2023, doi:10.3390/v15091972_

Round 1

Reviewer 1 Report

Authors described herein the genomic identification and characterization of four new viruses isolated from a Physostegia virginiana plant. However, sequences listed with their assession number were not available on NCBI database, also the first accession number is erroneous as it corresponds to the genome of an umbravirus. Otherwise, all other remarks and proposal are in the attached file.

Author Response

We would like to thank the respected reviewer 1 for your positive comments and valuable suggestions. We have carefully considered the comments and revised the manuscript accordingly. With these improvements, we hope that the current version can meet the journal’s standards for publication.

Reviewer 2 Report

The manuscript reports on the identification of viruses infecting Physostegia virginiana in China using diagnostic methods based on RNAseq HTS; HTS results were validated by means of (RT-)PCR with primer pairs designed on the genomes obtained, and phylogenetic analysis was performed on Sanger sequenced amplicons. Results indicated the presence of five viruses, one already well-known and characterized (AMV) and four novel viruses belonging to Fabavirus, Fijivirus and Caulimovirus genera. The Koch postulates were also verified for only one of these viruses, namely PVCaV1, in N. benthamiana.

The manuscript addresses an interesting and up to date topic with appropriate techniques, fitting figures and tables. Nonetheless, I suggest revisions before publishing, since some results are misleadingly reported and a series of experimental information is lacking. I call them minor revisions because they will not need further experiments, but they are substantial for the adequateness of the work to be published in an acceptable form. 

My concerns address the following specific points:

1) A huge amount of metadata is always provided along with HTS experiments which allow to evaluate the overall quality of the run and the assembly, i.e.: total number of reads, total number of contigs obtained (not only the "long" ones), quality of each contig (N50), k-mer(s) and other parameters used for assembly. All these data can also be provided as Supporting Information.

2) Some expressions in the manuscript as "contig amplification" should be amended in that they do not reflect what can really be done experimentally (see comments in the attached revision).

3) All analyses (RT-PCR and phylogenetic) regarding identified viruses of the Caulimovirus genus should be removed, because it is well known that they have the ability to integrate in the host's genome and are likely not to be infectious agents. The incessant identification of this genus in many hosts and the registration of such sequences in GenBank are in my opinion spoiling the literature and databases with useless data, impairing a correct use of HTS which is indeed a powerful technique, but still mostly relying on databases which should be as curated as possible.

4) Please fully explain how the 3000 bp amplicon obtained for PVCV2 was Sanger sequenced (see the comment in the attached revision).

Author Response

The manuscript reports on the identification of viruses infecting Physostegia virginiana in China using diagnostic methods based on RNAseq HTS; HTS results were validated by means of (RT-)PCR with primer pairs designed on the genomes obtained, and phylogenetic analysis was performed on Sanger sequenced amplicons. Results indicated the presence of five viruses, one already well-known and characterized (AMV) and four novel viruses belonging to Fabavirus, Fijivirus and Caulimovirus genera. The Koch postulates were also verified for only one of these viruses, namely PVCaV1, in N. benthamiana.

The manuscript addresses an interesting and up to date topic with appropriate techniques, fitting figures and tables. Nonetheless, I suggest revisions before publishing, since some results are misleadingly reported and a series of experimental information is lacking. I call them minor revisions because they will not need further experiments, but they are substantial for the adequateness of the work to be published in an acceptable form. 

Our response: We would like to thank the respected reviewer 2 for your positive comments and valuable suggestions. We have carefully considered the comments and revised the manuscript accordingly. With these improvements, we hope that the current version can meet the journal’s standards for publication.

My concerns address the following specific points:

1) A huge amount of metadata is always provided along with HTS experiments which allow to evaluate the overall quality of the run and the assembly, i.e.: total number of reads, total number of contigs obtained (not only the "long" ones), quality of each contig (N50), k-mer(s) and other parameters used for assembly. All these data can also be provided as Supporting Information.

Our response: We thank the reviewer for their questions. we have integrated these data into materials and results.

2) Some expressions in the manuscript as "contig amplification" should be amended in that they do not reflect what can really be done experimentally (see comments in the attached revision).

Our response: revised as suggested.

3) All analyses (RT-PCR and phylogenetic) regarding identified viruses of the Caulimovirus genus should be removed, because it is well known that they have the ability to integrate in the host's genome and are likely not to be infectious agents. The incessant identification of this genus in many hosts and the registration of such sequences in GenBank are in my opinion spoiling the literature and databases with useless data, impairing a correct use of HTS which is indeed a powerful technique, but still mostly relying on databases which should be as curated as possible.

Our response: We would like to reserve the results about these two new Calimoviruses. Sequence comparisons and phylogenetic analysis support these two calimoviruses were different with others that have reported, and were presumed to be new species of the genus Caulimovirus. The caulimovirus mainly replicates its DNA genome by reverse transcription of a pregenomic RNA, eventhough some cases of integration of caulimoviruses DNA with host genome is reported, not all caulimoviruses were able to integrated into the host DNA. Some caulimoviruses were found cause mottles or mosaics on vegetables, ornamentals, and weeds, which are accompanied by poor growth, poor quality, and reduced yields (George et. al. 2005). We can not rule out the possibility that the symptom of Physostegia virginiana samples might be caused by these two caulimoviruses infection.

4) Please fully explain how the 3000 bp amplicon obtained for PVCV2 was Sanger sequenced (see the comment in the attached revision).

Response: we sequenced amplicon longer than 1000 bp using further internal primers in sequencing company.
